



,
Wind Energy Science (wes)

# Segmented Gurney Flaps for Enhanced Wind Turbine Wake Recovery

Nirav Dangi[1,2], Koen Boorsma[1], Edwin Bot[1], Wim Bierbooms[2], and Wei Yu[2]

[1]Netherlands Organisation for Applied Scientific Research (TNO)
[2]Delft University of Technology (TU Delft)

**Correspondence:** Koen Boorsma (koen.boorsma@tno.nl)

**Abstract.** The wind turbine wake is a downstream region of velocity deficit, featuring higher turbulence and a complex helical vortex structure. In low ambient turbulence and low tip speed ratio conditions, the wind turbine wake is extremely stable. The impact of the velocity deficit is a power loss for the downstream wind turbine, which scales with the velocity cube. This study uses field tests and simulations to evaluate segmented Gurney flaps for enhanced wind turbine wake recovery and power output in a wind farm setting; where the power and loads of the retrofitted wind turbine were assessed. Four Gurney flaps were attached to each blade tip of a 3.8MW research wind turbine. This configuration is hypothesised (in line with an ECN (now TNO Wind Energy) patent) to cause a spatial variation to the stable tip vortex and induce turbulence in the wake for faster wake mixing. Field tests using a scanning LiDAR were conducted to quantify the wind turbine wake recovery between the baseline and the retrofitted configuration in various atmospheric conditions. The results show a consistent increase in wake recovery for the Gurney flap configuration, generally at all downstream distances (span wise averaged deficits reduced by roughly 10% at hub height, at a downstream distance of 5D), pronounced at low tip speed ratio conditions. Using crude assumptions, this implies a 4% relative increase in wind farm efficiency for a typical wind farm with outer rows of wind turbines with segmented Gurney flaps. The impact of retrofitting on turbine power and loads remained within the measurement uncertainty band, and this limited effect is confirmed by design load simulations In this work, a very successful field test campaign was executed which demonstrated the use of segmented Gurney Flaps as a promising add-on to promote enhanced wind turbine wake recovery for improved overall wind farm farm performance.

## 1 Introduction

The wind turbine wake is a region of three-dimensional turbulent flow characterized by a velocity deficit and a complex helical vortex structure. The wake of a single rotor blade consists of a continuous sheet of trailed vorticity due to the gradient in bound circulation along the blade span. In the wind turbine wake, a division can be made into the regions of near wake, intermediate wake, and far wake (Vermeer et al. (2003); Lignarolo (2016)). The near wake is taken as the area just behind the rotor where the presence of the wind turbine rotor is apparent by the number of blades, blade aerodynamics, including stalled flow, 3-D effects, and the tip and root-vortex helices. In the far wake the wake-generated turbulence and the external atmospheric



turbulence contribute to the breakdown and diffusion of the tip-vortex spiral and most of the turbulence mixing happens, while the wake undergoes a re-energising process. Between these two regions, a third zone can be distinguished, the intermediate wake (Lignarolo (2016)). This is the region where the turbulent mixing begins to prevail on the organized vortical structures, where the tip-vortex spirals may start to interact mutually and become unstable. The instability and breakdown of the helical system of vortices in the near wake affects the development of the turbulence in the far wake, where the mixing process between the inner and the outer flow regions occurs (Lignarolo (2016)). With low ambient turbulence and low tip speed ratio (rated and above rated wind speed conditions), the wind turbine wake is extremely stable. The velocity deficits in wind turbine wakes are critical for bigger wind farms, where multiple turbines are grouped together and their different wake effects are combined. The resulting energy loss of a wind farm can even be 20% for a farm of 140 turbines with a spacing of 5 rotor diameters (Schepers (2012)).

Researchers have proposed and investigated several wake control strategies to decrease the power loss of downstream wind turbines by steering (deflecting the wind turbine wake away from another downstream turbine) or weakening the upstream wakes (faster wind turbine wake recovery by inducing increased wake mixing or by pitching to lower angles of attack to cause change in thrust coefficient). Static axial induction control by pitch (Boorsma (2012); Dilip and Porté-Agel (2017); Wang et al. (2016)) or torque control (Gebraad et al. (2015); Bartl and Sætran (2016)), dynamic axial induction control (Westergaard (2012); Goit and Meyers (2015)) by dynamic pitch control (Frederik et al. (2020); Munters and Meyers (2018); Frederik et al. (2019)) usually fall into the latter category. Add-on devices focusing on faster wake mixing by use of actuating flaps (Huang et al. (2019); Zhang et al. (2020); Marten et al. (2020)), turbulators (Van Garrel and Bot (2014); Ceyhan Yilmaz and van Kalken (2017)) and winglets (Mühle et al. (2020); Ceyhan Yilmaz and van Kalken (2017)) on the wind turbine blade tip are also evaluated in existing research. Gurney flaps have been evaluated mainly for load control for wind turbines in some studies, for example, in Alber et al. (2020); Boorsma and Schepers (2016); Kentfield (1994). However, most of the studies focus on root Gurney flaps and thus operate in a different angle of attack regime than the tip Gurney flaps used in this study. There is little to no literature on the effects of tip Gurney flaps on wind turbine blades.

The Gurney flap, named after the race car driver Dan Gurney, is a simple small tab (height usually lower than 2% of the airfoil chord) added to the trailing edge (perpendicular to the free-stream) of the high-pressure side of an airfoil. They can increase the lift considerably with only a small drag penalty and also change the downstream wake development (Liebeck (1976)); which also depends on the height of the Gurney flap (Figure 102 in Jeffrey (1998)). There is plenty of existing research into the aerodynamics of Gurney flaps. In Wang et al. (2008), it was found that with the application of Gurney flaps, a long wake downstream of the flap containing a pair of counter-rotating vortices is formed. This can delay or eliminate the flow separation near the trailing edge on the upper surface, increasing the total suction, leading to an increased circulation with an enhanced lift. The Gurney flap provides a substantial increase in lift before the stall angle, after which the drag penalty causes a reduction in lift to drag ratio.

When utilising Gurney flaps for enhanced wake recovery or wake vortex alleviation, Matalanis and Eaton (2007, 2006); Delnero et al. (2016); Holst et al. (2016); Troolin et al. (2006) provide detailed study on the topic. In particular, segmented Gurney flaps have been evaluated on aircraft wings in the study of Matalanis and Eaton (2007), for wake vortex alleviation.





One of the configurations in Matalanis and Eaton (2007) is the closest to the segmented Gurney flap configuration used in this study on wind turbine blades, four along each blade tip. The expectations of wake recovery by use of segmented Gurney flaps were clarified with their study. They observe an intensification of tangential and stream-wise velocity components (Figure 3.13 in Matalanis and Eaton (2007)) from the segmented Gurney flaps. The reason behind it being the increase in circulation due to the change in the span-wise loading. The authors state that despite a smooth increment in the loading distribution in the configuration they experimented, a very small counter-rotating vortex pair also occurred at the flap tips which was then advected by the strong mean flow due to the primary trailing vortex. The velocity deficit around the Gurney flap is associated with the increased drag, however, as we go downstream it is no longer present; because as the vortex continues to roll up, the patches diffuse into one another by being constantly advected by the strong tangential velocities (which was also seen in study of Yang et al. (2020) and Holst et al. (2016)). The segmented Gurney flaps' effects were being felt by the vortex more and more as it continued to roll up. From the experiments, the authors established that in order to significantly perturb the vortex, only 13% of the span needed to be deployed with Gurney flaps. The flaps were applied only near the tip of the wing where the loading distribution varied the most. The insights on the intermediate wake from this study also confirmed that the effect of the segmented Gurney flaps upon the vortex was a lasting and reliable change. They also conclude that miniature trailing edge effectors (segmented Gurney flaps) can be used to introduce spatial disturbances to a trailing vortex in both the span-wise and lift directions. Finally, they suggest that the use of miniature trailing edge effector configurations (segmented Gurney flaps), which if varied in time, may be useful for wake alleviation. This study on an aircraft wing provides a detailed relevant literature for the use of segmented Gurney flaps for wake mitigation.

Some gaps in the above mentioned literature are the need of full scale field tests on wind turbines for further validation, concern for increased fatigue loading on the wind turbine on which the strategies are utilised, applicability at higher turbulence intensity levels and practicalities to add the devices on existing wind turbine blades. To overcome these gaps, this study evaluates segmented Gurney flaps for wind turbine blades with the following methodology:

1. Setting up and conducting Scanning LiDAR measurements in the wake (up to a distance of 5.5 times the rotor diameter downstream) for a 3.8MW research wind turbine with and without these segmented Gurney flaps. The segmented Gurney flaps (four on each blade, each 0.6m apart) are retrofitted to the 3.8MW research wind turbine.

2. Analysing wake recovery (time averaged profiles) of both configurations for different free stream wind speed, turbulence intensity, and wind direction conditions.

3. Performing simulations to qualitatively validate faster wake breakdown and quantitatively assess the power and loads of the retrofitted wind turbine.

The hypothesis in this study is in line with the ECN (now TNO Wind Energy) patent (Van Garrel and Bot (2014)). The hypothesis behind the use of segmented Gurney flaps is to alter the lift distribution along the blade span (at the tip); achieve a jagged lift (Figure 1) and circulation distribution which causes additional stronger vortices shed from the edges of Gurney flaps to perturb the stable tip vortex. This use of segmented Gurney flaps is hypothesised to cause a spatial disturbance to the





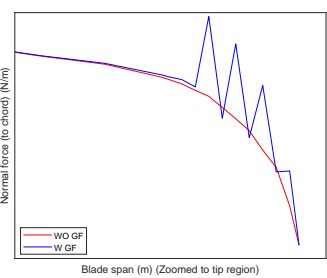

Figure 1: Normal force distribution without and with segmented Gurney flaps

tip vortex. Additionally, a higher pressure drag from the Gurney flap is hypothesised to potentially increase turbulence in the
wake which could contribute to the faster wake mixing.

The next chapter details the methodology employed in this study. Following that, the set up of the field tests is explained
in section 3. The results of wake analysis are discussed in subsection 4.1 and the results of the power and loads analysis are
briefly discussed in subsection 4.2. Finally, concluding remarks and recommendations are placed in section 5. Validation and
background information on certain parameters are provided in the Appendix.

## 2 Methodology

This study comprises of two methods to assess the impact of segmented Gurney flaps, which are, field tests and simulations
(Figure 3). The wind turbine under study was a 3.8MW research wind turbine with a rotor diameter of 130m and a hub height of
110m. The wind turbine is located at a wind farm in Netherlands (Figure 2). The field tests for the baseline configuration were
conducted from $01-09-2022$ to $23-01-2023$ and for the retrofitted configuration, from $24-01-2023$ to $14-02-2023$.
See subsection 4.1 for remarks about the shorter testing period for the retrofitted wind turbine.

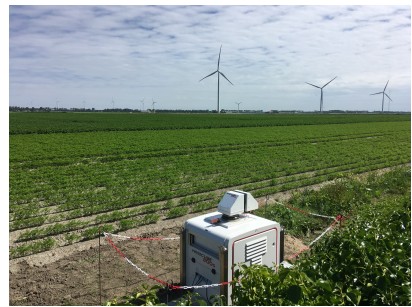

Figure 2: Test site (Wieringermeer, The Netherlands) (Photographed by TNO)

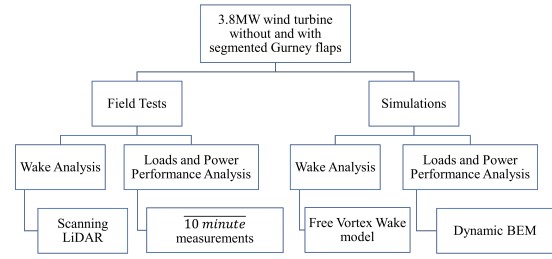

Figure 3: Methodology





The retrofitting of the above mentioned wind turbine was done by installing four Gurney flaps on each wind turbine blade tip region. The Gurney flaps were designed as a wedge and had a height and width of 2% chord and 0.6m length. The same height and width ensured a 45 degrees wedge and in line with literature, these dimensions were chosen to ensure a better lift to drag ratio than the typical rectangular Gurney flap (Bloy and Tsioumanis (1997); Hao and Gao (2019)). Figure 4 below illustrates

the dimensions:

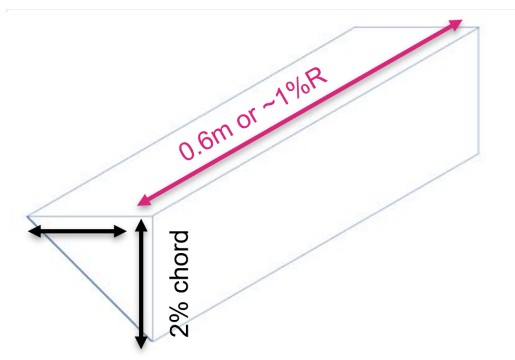

Figure 4: Gurney flap designed as a wedge

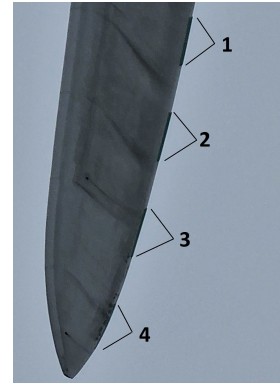

Figure 5: Installed Gurney flaps (Photographed by TNO)

In Figure 5 the Gurney flaps as installed on the wind turbine blade are labelled. They are defined as segmented Gurney flaps because of the spacing between them, as also shown below in Figure 6:

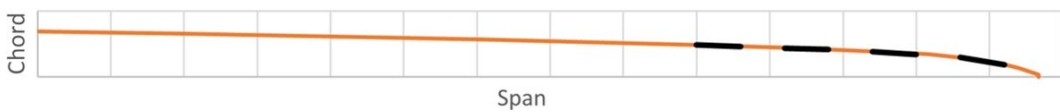

Figure 6: Gurney flaps (Blue thick line) shown on wind turbine blade (only a part of the blade is shown)

The spacing between the Gurney flaps as seen above, is introduced to impart the spatial variation to the stable tip vortex, in line with the hypothesis explained in section 1. The Gurney flaps were manufactured with SikaBlock M940, which has a

density of 1200kg/m$^3$. The resultant weight came to a maximum of 130g for one Gurney flap. The manufactured Gurney flaps were installed to the wind turbine blade by technicians on a cherry picker, by use of Plexus MA 320 adhesive.

For the wake analysis in field tests, a Leosphere Vaisala windcube 200S pulsed Scanning LiDAR was used. To assess the inflow conditions, a ground based profiling LiDAR was used, which was placed $\approx 2.2D$ upstream ($\approx 282m$). This ground based profiling LiDAR was used to assess the inflow conditions at 11 different altitudes between 42m and 188m. Free vortex

wake module of NREL OLAF (Shaler et al. (2020)) was used to further analyse the impact of segmented Gurney flaps on the wind turbine wake.





For the power and loads analysis in field tests, ten minute averaged measurement data was utilised. However, the short testing period caused large standard errors of turbine power and loads data. These standard errors were in the order of the relative differences of the results. Thus, because of this high uncertainty, the power and loads analysis of field tests is not presented
here (interested readers may refer Dangi (2023)). To estimate the power and loads in simulations, dynamic blade element momentum theory (DBEM) was used from NREL OpenFAST (Jonkman et al. (2022)), which is OpenFAST terminology for blade element momentum theory with dynamic inflow model.

Lastly, in regards to the generation of the airfoil polars for the airfoil with the Gurney flap design as shown in Figure 4, a 2-D computational fluid dynamics (CFD) study with Reynolds-averaged Navier–Stokes (RANS) modelling was conducted with a
130 standard setup. The $k - \omega$ SST turbulence model was used with a Y-plus value of $0.1$, with a O-grid. Prior to the simulation of airfoil polars with Gurney flaps, the CFD set-up was validated by comparing to wind tunnel data from the airfoils without Gurney flaps.

## 3 Setup and Data Processing of Field Test Wake Measurements

The undisturbed wind sector for the wind turbine under study was $180°$ to $340°$ (South to North- Northwest wind directions),
measured with respect to North. This is shown in Figure 7 below, where the blue lines indicate the undisturbed sector.

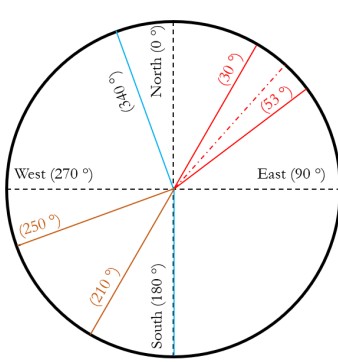

Figure 7: Compass plot

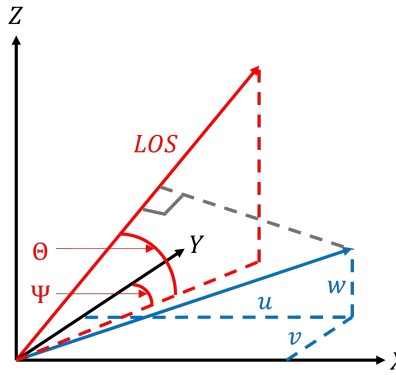

Figure 8: Scanning LiDAR co-ordinate system

The brown lines in Figure 7 indicate two wind direction limits which capture the prevailing wind sector at the test site. The Scanning LiDAR was placed $\approx 912$m upstream of the wind turbine, and the scan settings utilised are shown in Table 1. The azimuthal ($\psi$) ranges of these settings are depicted as red lines in Figure 7. The red center line represents the approximate axis of the wind turbine (for $230°$ wind direction with zero yaw misalignment), and the scan pattern was made symmetric about
140 it. The scanning LiDAR azimuth ($\psi$) and elevation ($\theta$) convention is shown in Figure 8. LOS implies the line of sight of the LiDAR. The scan time was $\approx 2.8$ minutes, this implies that in a ten minute interval, roughly three samples were available at every point of the scan.





Table 1: Scanning LiDAR settings (Wind turbine is located at $\approx 41.5°$ azimuth)

| Parameter | Min. | Max. | Step size | No. of unique points | Time for 1 scan (minutes) | Total no. of points in ideal scan |
|---|---|---|---|---|---|---|
| $\psi$ (°) | 30 | 52.88 | 0.22 | 105 | | |
| $\theta$ (°) | 4 | 7.5 | 0.5 | 7 | 2.8 | 22050 |
| Range ($m$) | 900 | 1625 | 25 | 30 | | |

The field test wake analysis results presented in subsection 4.1 are for the wind sectors of 230° to 250° (Southwest to West-Southwest). This sector allows for most samples given the prevailing wind sector, and has (theoretically) less uncertainty in retrieving wind components from the LiDAR output of line of sight or radial wind speed. This is because in this sector the LiDAR azimuth ranges and the free stream wind direction are well aligned. This sector is also chosen because it allows for wake capture up to $\approx 5D$ downstream. Furthermore, the field test wake analysis results of the baseline configuration are presented for the period from $25-11-2022$ to $23-01-2023$ and all the measurements conducted on the retrofitted configuration from $24-01-2023$ to $14-02-2023$. This choice for the baseline configuration measurement was based on the atmospheric stability conditions. During these winter months, it was found that the prevailing atmospheric stability conditions were either neutral, stable or very stable and the unstable conditions were hardly present. This was concluded on the basis of a Bulk- Richardson method (Grachev and Fairall (1997)) atmospheric stability analysis using bins established in Gryning et al. (2007). In this way, wake analysis comparisons were done during similar atmospheric stability conditions as per the bins.

LiDAR measurements of the wind field work through analysis of laser light reflected from aerosol particles in the air. The frequency of the back scattered laser light is shifted from its original value, due to the Doppler effect, in proportion to the speed of the reflecting particle resolved along the direction of the laser beam and the radial wind speed $V_r$. In general, aerosol backscatter, relative humidity, precipitation, and atmospheric refractive turbulence affect the LiDAR performance (Aitken et al. (2012)). A set of possibly faulty data is usually reflected in the carrier to noise ratio, provided by the LiDAR. The steps taken for the data processing of the scanning LiDAR data are shown in Figure 9. Firstly, a carrier to noise ratio filter was used such that data within the range of $-23$dB and $-3$dB was preserved (Cassamo et al. (2021)).

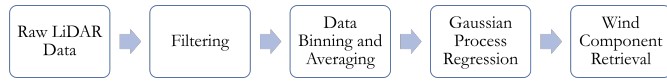

Figure 9: Scanning LiDAR data post processing

As the scope of this study was to assess the mean wake profiles, rather than high frequency measurements, a data binning and averaging step was taken. The data binning was quantified by different inflow conditions of wind direction, wind speed





and turbulence intensity levels. As mentioned previously, a ground based LiDAR was used to assess the inflow conditions for quantification of the wake analysis in various bins. The parameters of data binning are listed in Table 2 below and the ten minute averaged measurements of these parameters were used.

Table 2: Data binning of Scanning LiDAR data

| Parameter (at hub height) | Bin limits |
|---|---|
| Turbulence intensity (TI in %) | 0% to 5%, 5% to 8.5%, and 8.5% to 11% |
| Wind direction (°) | 220° to 250° |
| Wind speed (m/s) | 6m/s to 11m/s in steps of 1m/s |

For the turbulence intensity, the relation of $TI = \frac{\sigma_U}{\overline{U}}$ was used with the standard deviation and the average value of the inflow wind speed for a ten minute period. The hub height parameter based filtering might not be adequate at all times. For example, since the ten minute averaged measurements were utilised for the data bins, it could be that during a period with gust, the average value is not fully representative of the wind turbine performance during the ten minute interval. Therefore, two additional steps were taken. The power law wind shear exponent and the wind turbine aerodynamic performance were assessed. The power law wind shear exponent was ensured to be within $\pm 0.1$ for the comparison of the two configurations. The latter, aerodynamic performance check was analysed with respected to rotor power, rotor speed and blade pitch values. If the variation of these parameters was found to not be within acceptable ranges then the corresponding sample was filtered out.

With the filtered data set, a complete wind field visualisation according to the scan settings listed in Table 1 was found to be lacking during certain instances. This was tackled by utilising Gaussian Process Regression on the bin averaged data set. The ARD (Automatic relevance determination) Matern 3/2 Kernel was utilised in this work using the 'fitrgp' function of MATLAB (Mathworks ‖ MATLAB (2022)). The reason of this choice is that the Matérn 3/2 kernel is a standard kernel for environmental data and that upon testing of different kernels on the data-set, the ARD Matérn 3/2 kernel resulted in a good match with observations. Gaussian process regression has been utilised in previous research (Stock-Williams et al. (2018); Cassamo et al. (2021)), for high frequency LiDAR measurements. The use of Gaussian process regression helped to smooth the data set (in addition to the reliable interpolation at gaps in data), given the inherent standard error in the bin averaged scanning LiDAR data set.

Finally, the wind component retrieval step was taken since a LiDAR only provides the radial component of wind speed. The choice of sectors (not affected by nearby wakes) makes the retrieval of the necessary wind components less challenging and less prone to error. Using geometric relations (and assuming the LiDAR probe volume to be negligible) the following equation was used to write the radial velocity in terms of the $u, v, w$ components and the scanning LiDAR's azimuth ($\psi$) and elevation ($\theta$) as follows:

$$V_r = u\sin\psi\cos\theta + v\cos\psi\cos\theta + w\sin\theta \tag{1}$$





This equation follows from Figure 8, where the sign convention followed is that the $0°$ azimuth points to the North and $90°$
East and so on. The elevation is $0°$ in the horizon and increases towards zenith in the vertical plane. Furthermore, the horizontal
wind speed $V_h$ and the wind direction $\alpha$ can be determined as follows:

$$V_h = \sqrt{u^2 + v^2} \tag{2}$$

$$\alpha = \arctan \frac{u}{v} \tag{3}$$

As seen in Equation 1, there are three unknowns ( $u, v, w$) but only one equation. This is commonly known as the Cyclop's
dilemma (Simley and Pao (2012)). Two assumptions were made to solve the equation. Firstly, the $w$ component (vertical) was
assumed to be zero, as it can be considered negligible (very low elevation angles (Table 1), lead to a small sine component).
Secondly, it was assumed that the wind direction is homogeneous throughout the wind turbine wake. This wind direction was
assumed as the inflow wind direction at the hub height of the wind turbine under study, $\approx 2.2D$ upstream. These assumptions
lead to the following relations:

$$V_h = \frac{V_r}{\cos(\overline{\alpha} - 180 - \psi)} \tag{4}$$

In the above equation, $\overline{\alpha}$ represents the mean wind direction of the incoming wind. This approach was compared against
the commonly used non linear least square fitting method (Bodini et al. (2017); Peña et al. (2019)) and a slightly sophisticated
Maximum A Posteriori method (by using UQLab (Marelli and Sudret)), provided in section B.

With this discussion, the data processing steps employed for the field test wake measurements are explained. For the results
presented in subsection 4.1; first, the inflow conditions are shown for the respective wind speed bins. As mentioned in Table 2
the hub height parameters were the first binning criteria, so the inflow profiles through the various altitudes (as measured by
the ground based profiling LiDAR) are shown to realise that certain differences in the wake comparison could be present due
to the wind speed and turbulence intensity shear. The results are presented for the hub height contours, wake profiles in the
axial (along wind direction) and vertical (along altitude) planes.

The hub height contours were made by combining the points $\pm3$m from the hub height of 110m, in accordance with the
scanning pattern (Figure A2). The axial wake profiles are shown as a span wise average at different downstream distances. This
span wise average was calculated on basis of a wake width of 1.5D for both configuration; $1D$ wake width was based on the
rotor end points at the corresponding yaw angles and a 0.25D was added on both sides of this wake width to roughly allow the
width to take into account wind turbine stream tube expansion. The wake width and the rotor hub line is indicated in the hub
height contours with green asterisks. These profiles are shown for four different altitudes.

The vertical wake profiles at different altitudes and downstream distances are shown for the $+50\%$ blade span location in
the lateral direction (example, Figure B1), with the assumed wake width as discussed above. The $+50\%$ span location was
chosen to ensure proper wake capture in all results, upto 5D downstream. Ideally these would have been plotted at the $\pm75\%$
span line, under the assumption that this point will correspond to the maximum lift and thus would roughly be the point of
highest deficits (Vermeer et al. (2003); Aitken et al. (2014)). However, sometimes this point was found to lie away from the
scan pattern or would coincide with the line along the nacelle which was poorly resolved by the LiDAR (for example, the spike





in wind speed along the nacelle in Figure 11(a)). To avoid erroneous profiles, the $+50\%$ span was chosen and it also ensures uniformity throughout the wind speed bins. Also note that the highest deficit points would likely be different for the retrofitted

wind turbine.

## 4   Results

### 4.1   Wake Analysis in Various Inflow Conditions

In this section, first the field test wake analysis results are discussed. Following that, the simulation results are briefly discussed. Standard error patches are used to indicate the uncertainty in the field test results. The standard error was defined as $\frac{\sigma}{\sqrt{N_{samples}}}$,

that is, the ratio of the standard deviation of the measurements divided by the square root of the number of samples. Similar trends were observed in the five wind speed bins shown in Table 2, so, only two wind speed bins are discussed here, which are corresponding to inflow conditions of 8m/s-9m/s and 10m/s-11m/s. The choice of bins is based on the fact that the latter conditions are close to the rated wind speed of the wind turbine, which relates to operational conditions such that the tip speed ratio and thrust coefficient are low (but roughly maximum thrust in dimensional sense) which leads to more stable and longer

baseline wind turbine wake. The former falls at a higher tip speed ratio and thrust coefficient which leads to a less stable and shorter baseline wind turbine wake. This choice allows visualisation at varied operational conditions of the 3.8MW research wind turbine. The number of scans averaged to present the results are indicated on the contour plots and holds for axial and vertical profiles as well. For example, if the number indicated is 12 scans, then the wake measurements were taken for four 10 minute intervals (in line with the scan time shown in Table 1).

First, the results of the wind speed bin of 8m/s to 9m/s are discussed. The figure below illustrates the vertical profiles of the incoming wind speed and turbulence intensity:

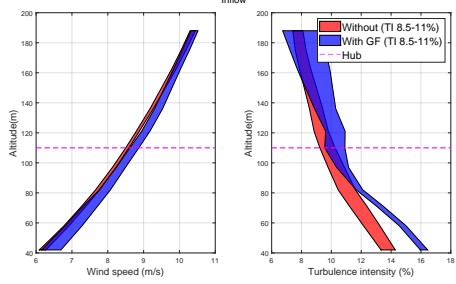

Figure 10: Inflow profile (8m/s$< U_{\infty,hub} < $9m/s)

From Figure 10, it should be noted that the inflow turbulence intensity at the lower altitudes is higher by roughly $0.5\%$ for the retrofitted configuration. With regards to the discrepancy in wind speed, the wake analysis results are presented for the normalised deficit values to avoid bias because of the different wind speed distribution in the bin averaged data. The hub height

contour below shows the normalised deficit profiles for the baseline and the retrofitted configuration:





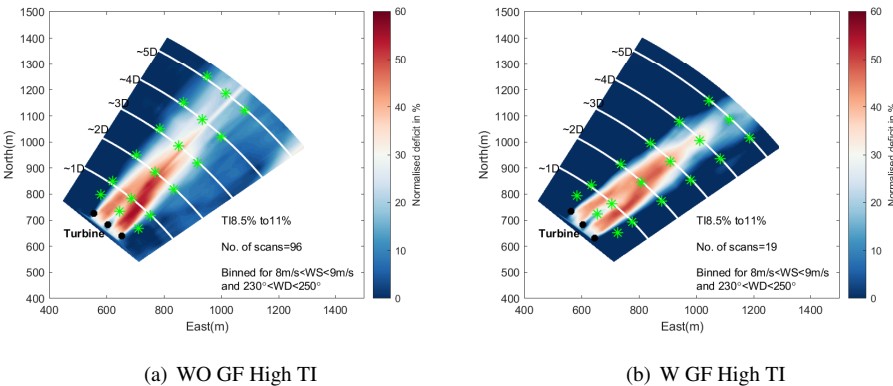

(a) WO GF High TI

(b) W GF High TI

Figure 11: Wake visualisation at hub height (8m/s$< U_{\infty,hub} < $9m/s) (Green asterisks represent the assumed 1.5D wake width and rotor hub line)

This wind speed bin corresponds to a high wind turbine tip speed ratio and a high thrust coefficient leading to a higher turbulence in the wake. In line with this operating condition, due to the high induction factor, a high deficit, about 60% is seen in the wind turbine wake, close to the Betz limit of roughly 67%. Such observations have also been made in previous studies (Aitken et al. (2014); van Dorp (2016)). A first observation that can be made is that the wake in the retrofitted configuration
appears to be thinner than the baseline configuration. Whether or not this is attributed to the addition of the segmented Gurney flaps is unclear because this observation was not consistent in all wind speed bins. The span wise averaged axial wake profiles shown below provide more insight into the wind turbine wake of the two configurations:

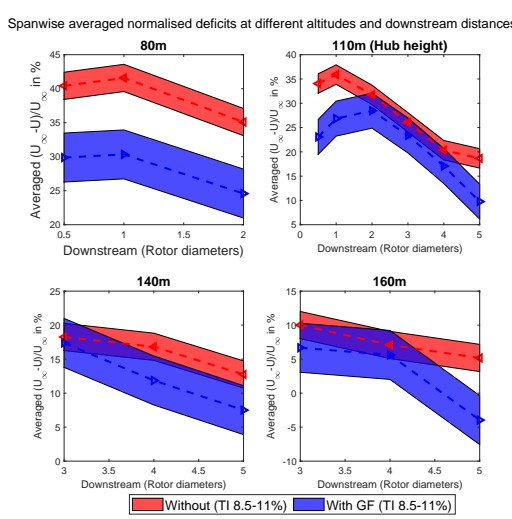

Figure 12: Averaged axial (along wind) wake profiles (8m/s$< U_{\infty,hub} < $9m/s) (Normalised with $U_\infty$ at hub height)

 

The axial profiles indicate a faster recovery in the retrofitted configuration at the different altitudes and downstream distances. The peaks for the deficit appear around 1D downstream after which both configurations show recovery. This peak deficit is expected because of the relaxation of pressure gradients (Ainslie (1988); Sanderse (2009)). Further downstream, the turbulent mixing prevails and the wake recovery is influenced by the ambient turbulence. However, the retrofitted configuration indicates a faster wake recovery also in proximity to the rotor, although with higher standard errors. This indicates an increase in turbulence in the near wake due to the segmented Gurney flaps. From the literature provided in section 1, the cause for the enhanced wake recovery in this region could be linked to the increased tangential velocity from the segmented Gurney flaps and the additional pressure drag associated with it. However, this study employed only one scanning LiDAR which was aligned with the prevailing wind direction, thus, the tangential component could not be assessed reliably. The vertical profiles below also indicate the same trend at all downstream distances:

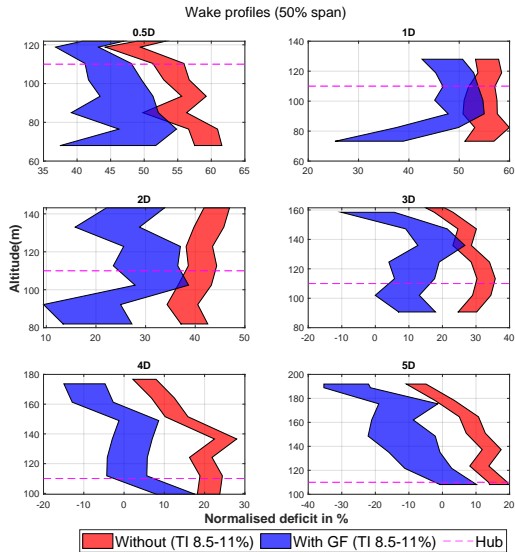

Figure 13: Vertical profile (8m/s$< U_{\infty,hub} <$9m/s) (Normalised with $U_{\infty}$ at hub height)

In the 0.5D downstream region, the wake profile shapes for the two configurations are similar, with lower deficits for the retrofitted configuration. In 1D and 2D downstream region, there appears to be an indication of a more steep slope in the wake of the retrofitted configuration and a trend for a double Gaussian wake profile. Overall, lower deficits are observed at the compromise of the lower number of scans in the retrofitted configuration leading to higher standard errors.

Next, the 10m/s to 11m/s wind speed bin results are discussed. The inflow conditions are shown below. The wind speed and turbulence difference are within the bin width for both the configurations, but a slightly higher value of turbulence is evident for the retrofitted configuration.

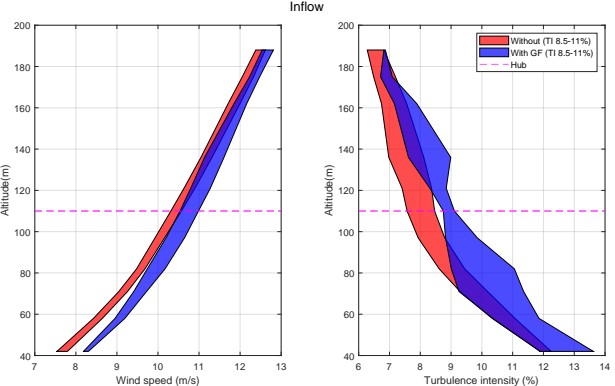

Figure 14: Inflow profile (10m/s$< U_{\infty,hub} <$ 11m/s)

Upon checking the rotor aerodynamic parameters as discussed in section 3, a 6% increase in power production was obtained for the retrofitted configuration, partly attributed to the slightly higher wind speed and mainly attributed to the addition of the segmented Gurney flaps. This power increase could possibly have a negative effect on retrofitted configuration wake recovery, because the wind turbine extracted more energy from the incoming wind. In this wind speed bin, the wind turbine operates at the highest thrust in dimensional form but a lower thrust coefficient and tip speed ratio than the previous wind speed bin. The

effect of this condition gets reflected in the hub height contour below. As expected, at such operating conditions the turbulence in the near wake would be limited (Troldborg (2009); Martínez-Tossas et al. (2022)) leading to a much longer wake as clearly seen in Figure 15(a):

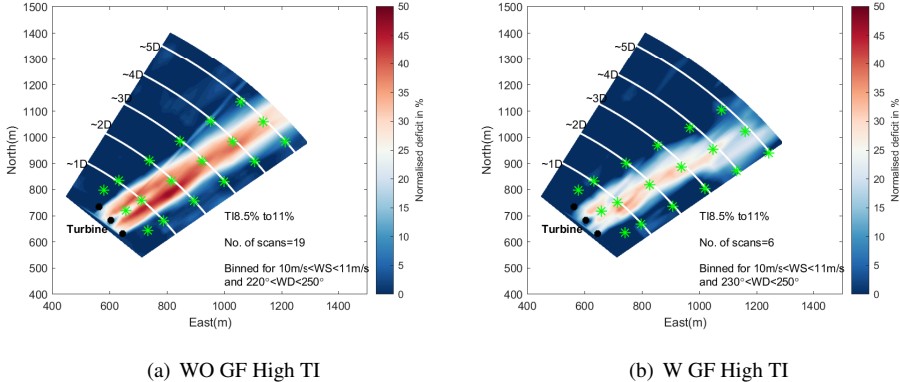

(a) WO GF High TI

(b) W GF High TI

Figure 15: Wake visualisation at hub height (10m/s$< U_{\infty,hub} <$ 11m/s) (Green asterisks represent the assumed 1.5D wake width and rotor hub line)

Now in comparison to the previous wind speed bin, the deficits in this wind speed bin are lower than the 8m/s to 9m/s wind speed bin; due to the lower thrust coefficient. From Figure 15(b), a chaotic wake is seen in the retrofitted configuration; likely





due to inadequate averaging and potentially an increased turbulence in the wake upon addition of segmented Gurney flaps. The span wise averaged axial wake profiles are shown below:

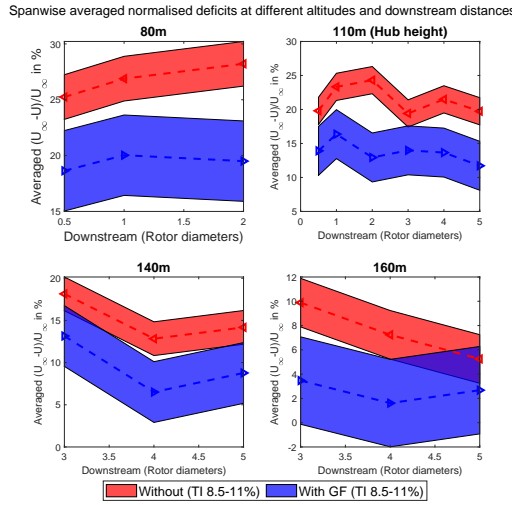

Figure 16: Averaged axial (along wind) wake profiles (10m/s$< U_{\infty,hub} < 11$m/s) (Normalised with $U_\infty$ at hub height)

Once again, the enhanced wake recovery is observed in the retrofitted configuration at all heights and downstream distances. The extremely stable structure of the tip vortex in the operating condition in this wind speed bin would imply a pronounced effect of the segmented Gurney flaps. This is also seen in the vertical profiles below:

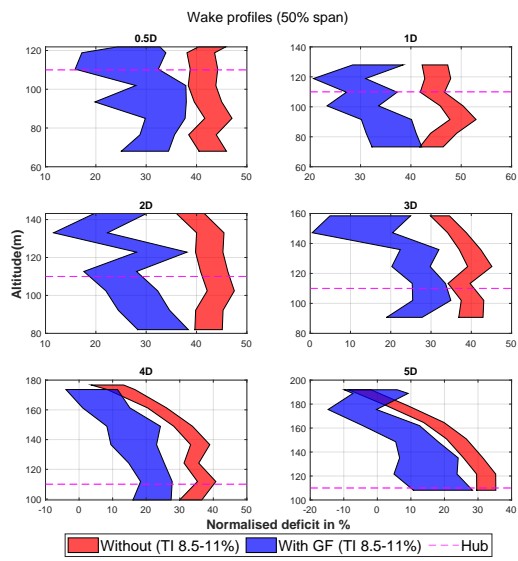

Figure 17: Vertical profile (10m/s$< U_{\infty,hub} < 11$m/s) (Normalised with $U_\infty$ at hub height)





The increased wake recovery is observed in proximity to the rotor in this wind speed bin as well. It should be noted that the number of scans for the retrofitted configuration are only 6, which means wake measurements during two 10 minute intervals were used.

The shorter testing of the retrofitted configuration was because of the increase in perceived noise levels of the retrofitted wind turbine. The sharp profile of the segmented Gurney flaps designed as shown in Figure 4 was expected to cause an increase in
noise level. Further design proposals were made, but due to time restrictions a new design could not be implemented in a practical manner. This lead to the segmented Gurney flaps being removed earlier than desired. The noise levels were not technically measured, so they cannot be quantified here. Overall, great insights into the wind turbine wake were made possible with the field tests conducted in this study. A reduction of the velocity deficit of 10% at 5D downstream distance translates into a 30% reduced wake loss. However, since the effect is only severe for the wakes from wind turbines in low turbulent free
stream, while the distance between turbines in large wind farms is generally much larger than 5D, the overall effect for a wind farm will be much less, probably around 3% to 5% relative reduction of the wake losses. However, dedicated farm simulation are recommended to verify this. In addition to that, the application of segmented Gurney flaps could result in an adjusted farm lay-out with closer spacing, yielding a higher energy density.

The free vortex wake simulations indicate that the retrofitted configuration has significant disturbances in the rotor tip
region, from roughly 2.5D downstream. While for the baseline configuration such significant disturbance is seen only after 5D downstream, as shown below:

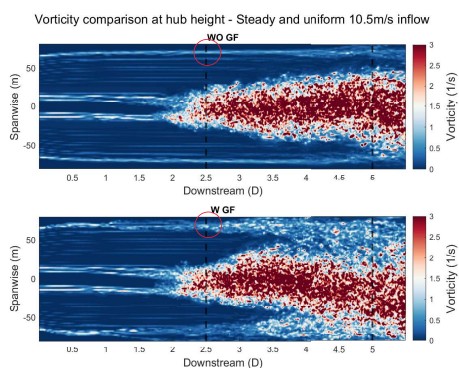

Figure 18: Wake mean vorticity visualisation at hub height 110m

This faster (earlier) mutual interaction of the tip vortices provides validation for one of the reasons behind the faster wake recovery by segmented Gurney flaps in the field tests, specially in farther downstream distances. The field test results also indicate the faster wake recovery in proximity to the rotor. This could not be validated in the simulations because the actual
Gurney flap was not resolved in the simulations, and only the airfoil polars were changed. This validation may be made possible by longer field tests or blade resolved simulations. Segmented Gurney flaps do show promising results on wake recovery, which should be validated by future longer field tests; recommendations for which are listed in section 5.



## 4.2 Power and Loads Analysis in Various Inflow Conditions

The field test measurements of the retrofitted turbine power and loads were associated with large standard errors due to limited
data. Thus, conclusive arguments about the impact of the segmented Gurney flaps could not be made with this data. So, simulations were set up in accordance with IEC normal turbulence model (NTM) and the cut in to cut out wind speeds were simulated using TurbSim (Jonkman and Kilcher (2012)). The mean angle of attack on the wind turbine blade tip was assessed. The angle of attack ranged from roughly 2 degrees to 0 degrees for $4 - 10$m/s wind speed range and then decreasing to roughly $-8$ degrees to $-12$ degrees for $24 - 25$m/s wind speed range. The range of angle of attack follows for the span wise positions
corresponding to the Gurney flap 1 to 4 (Figure 5). Because of the small spanwise length of the Gurney flaps, the integral of the normal force (to chord) curve increased by maximum 2%. The increase in annual energy production of the retrofitted wind turbine was found to be roughly +0.2%. The damage equivalent flap wise bending moment was found to increase slightly with a maximum of +5% around the partial load region of the wind turbine.

## 5 Conclusions and Recommendations

The aim of this study was to evaluate the use of segmented Gurney flaps on wind turbine blade tip to enhance the wind turbine wake recovery and assess the impact on the retrofitted wind turbine performance.

The main hypothesis behind the use of segmented Gurney flaps for faster wake recovery was to impart a spatial disturbance to the tip vortex. This would be imparted by the resulting jagged lift and circulation distribution on the wind turbine blade tip, which leads to additional vortices shed from the edges of the Gurney flap resulting in earlier mutual inductance of the vortex
filaments. Additionally, the pressure drag due to the Gurney flap was hypothesised to increase turbulence in the wake. Four Gurney flaps (as a wedge) were attached to each blade of a 3.8MW research wind turbine.

Field measurements using a scanning LiDAR, were quantified with bins of wind direction, wind speed and turbulence intensity; by measurements from a ground based profiling LiDAR. The scanning LiDAR was used to scan a sector upto $5D$ downstream at different altitudes in a a sector which was approximately 20 degrees wide. The measurements were binned and
averaged for wake analysis. Upon binning, Gaussian process regression was utilised to enhance the wind field visualisation. This approach allowed for reliable interpolation at gaps in data and smooth the bin averaged LiDAR data with the inherent standard error. Homogeneity of wind direction assumption was utilised to retrieve the wind component from the LiDAR LOS velocity. This method was found to perform better than the non linear least square fitting approach and Maximum A Posteriori approach.

Overall, field tests were successfully conducted enabling great insights into the wind turbine wake in various conditions. Enhanced wake recovery was observed for the retrofitted configuration, generally at all downstream distances and altitudes; and even more at low tip speed ratio conditions. The retrofitted configuration results indicate lower span wise averaged deficits by roughly 10% at hub height, at a downstream distance of 5D; which would imply using crude assumptions a 4% relative increase in wind farm efficiency for a typical wind farm with outer rows of wind turbines with segmented Gurney flaps. Dedicated farm
simulation are recommended to verify this estimate to further promulgate this research. The retrofitted configuration results





were associated with higher standard error because of the shorter testing period, which was a result of the higher noise levels from the Gurney flaps so that they had to be removed earlier. The results indicate increased turbulence in the wake because of the faster wake recovery observed also in proximity to the rotor.

The hypothesis for the spatial disturbance to the tip vortex was verified by means of free vortex wake simulations conducted in steady and uniform inflow conditions. A considerably faster (earlier by roughly 2D) tip vortex breakdown was observed for the retrofitted configuration. The power and loads analysis was conducted by means of simulations using blade element momentum theory in IEC NTM inflow conditions, from cut in to cut out wind speed. The results indicate an AEP increase of roughly $0.2\%$ with increase in the damage equivalent flapwise bending moment at most by roughly $5\%$, around the partial load region of the wind turbine.

This makes segmented Gurney flaps a promising add-on to wind turbine blades for enhanced wake recovery. Recommendations to further promulgate the use of Gurney flaps are in regards to future field tests and future research on these devices. Longer field tests, around 1.5 months for each configuration will allow for further validation of the enhanced wake recovery. LiDAR will likely have to be used to assess the seeming increase in turbulence in the near wake, because blade resolved CFD and in-situ measurements in the wind turbine wake will not be practical. The power and loads of the downstream wind turbine

could be assessed in a future study to further quantify the results. Future research could incorporate highly resolved actuator line modelling Large eddy simulations (with optimal Gaussian kernel width) to assess the velocity profiles with various wind speed and turbulence intensity conditions. The time step requirement for such a study was found to be in order of micro-seconds and was not in the scope of this study. Such a study could also be utilised to further optimise the design and spacing of the segmented Gurney flaps on the blade as the velocity profiles can be assessed in such simulations. Finally, for operation on land,

the noise of the devices will have to be reduced. This is however not very relevant for off-shore operation.

## Appendix A: Scanning LiDAR Pattern

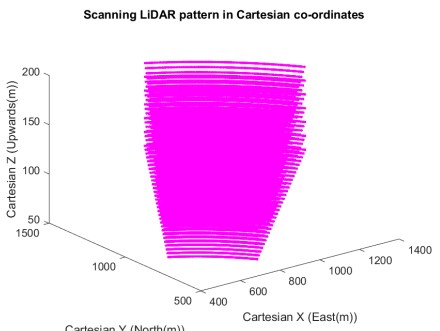

Figure A1: Scanning LiDAR pattern (Cartesian co-ordinates)

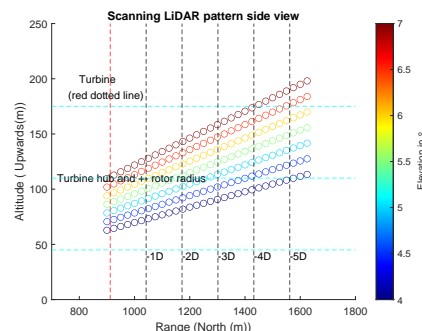

Figure A2: Scanning LiDAR pattern side view



## Appendix B: Wind Component Retrieval

For the non linear squares method, the Equation 1 was solved with $w = 0$ and the Levenburg-Marquardt algorithm was used to fit the $u$ and $v$ components, given the radial wind speed ($V_r$) and the scanning LiDAR azimuth ($\psi$) and elevation ($\theta$) in a non linear least squares sense. The Maximum A Posteriori method is a Bayesian estimation which incorporates prior belief about the unknowns, and then the posterior distribution of the unknowns is updated. This method is equivalent to the commonly used Maximum Likelihood estimation (MLE) when the said priors are uniform, that is, equal distribution of probabilities. For implementing MAP, Weibull priors were specified for $u, v$ components in the wind sectors of $190°$ to $250°$ sector by ground based profiling LiDAR measurements at hub height. The results indicate unrealistic estimates from non linear least square fitting method, specially on the edges of the scan (Figure 1(a)). The MAP method did not suffer from this issue but was computationally expensive. Based on the results of the comparison, the above mentioned wind direction assumption approach is used for the results presented in this study for its computational ease and applicability in the wake sectors analysed in this study. Furthermore, the results of the different methods at a $50\%$ span location (Figure B2) revealed similar trend with slightly different magnitudes.

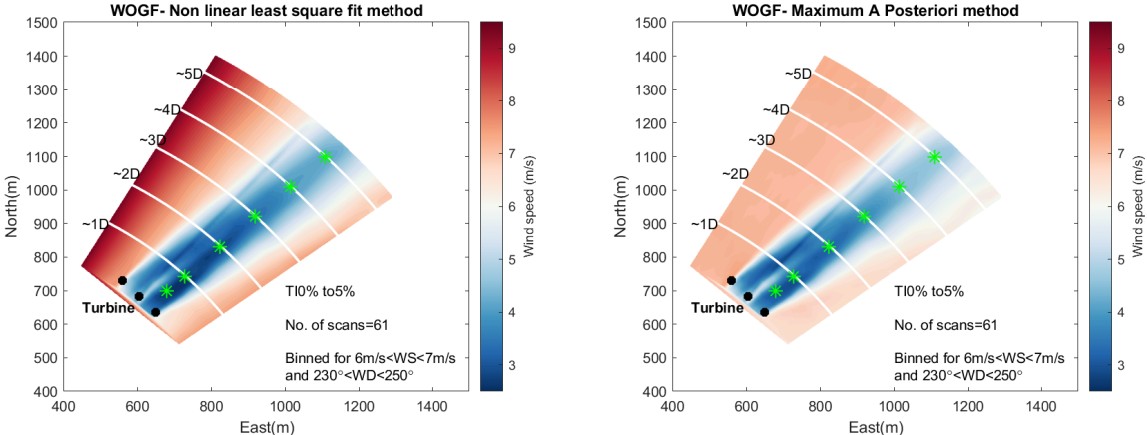

(a) Non linear least square fitting comparison (note the higher than inflow wind speeds, particularly on the left edge of the scan)

(b) Maximum A Posteriori estimation comparison (OK, but computationally expensive)

Figure B1: Horizontal wind speed ($\sqrt{u^2 + v^2}$) example for a bin of ($6m/s < U_{\infty,hub} < 7m/s$) [Green asterisks: 50% span location]

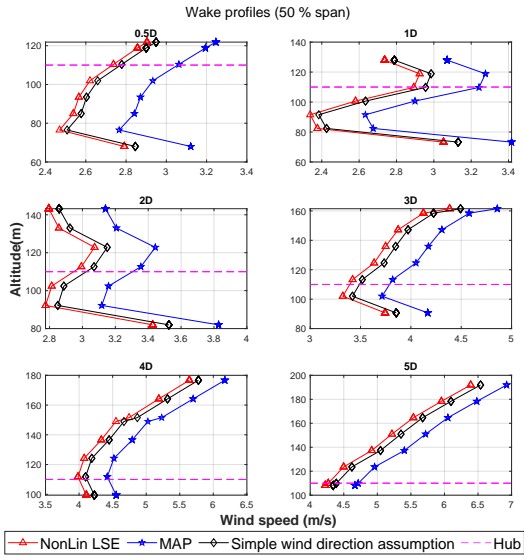

Figure B2: Vertical profile comparison of different methods of wind component retrieval ($6m/s < U_{\infty,hub} < 7m/s$)

*Code and data availability.* The code used to analyze the wake measurements is available upon request. Turbine model data and measurements are not cleared for public use by the manufacturer.

*Author contributions.* ND conducted the work presented here as a part of his master thesis. KB and EB acted as supervisors from TNO. WB and WY acted as supervisors from TU Delft. KB was heavily involved with the project management, conduction of field tests. EB proposed the first design and spacing of the segmented Gurney flaps as a wedge. With the excellent supervision from KB, EB, WB, WY; insightful 380 discussions were held to refine the work. Everyone reviewed the manuscript.

*Competing interests.* The authors declare that they have no competing interests.

*Acknowledgements.* The work conducted in this study is part of the TIADE (Turbine Improvements for Additional Energy). A consortium of TNO (Netherlands Organisation for Applied Scientific Research), GE Renewable Energy and LM Wind Power are collaborating on the TIADE project to develop technologies and design methods for more efficient operation of next-generation wind turbine rotors, wind farms 385 with large rotor wakes and demonstrate them in the field. TIADE has been co-financed with Topsector Energiesubsidie from the Dutch Ministry of Economic Affairs under grant no. TEHE119018.



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
