# Peer review of "Segmented Gurney Flaps for Enhanced Wind Turbine Wake Recovery"

_Wind Energy Science, 2023_

## Referee Comment (RC2)

**Overview:**

The manuscript entitled, "Segmented Gurney Flaps for Enhanced Wind Turbine Wake Recovery" by Nirav Dangi, Koen Boorsma, Edwin Bot, Wim Bierbooms, and Wei Yu endeavors to describe differences in wind, turbine week aerodynamics introduced by the presence of a gurney flaps. On the trailing edges of wind turbine Blade tips. There is a long history of flaps as passive or active control, mechanisms for wind, turbines, and a great deal of debate in the literature as to their merit. The authors are strongly encouraged to review this literature and contextualize their work within the spotty before presenting this for publication again. Overall the manuscript reads like a section of a thesis. By itself, the study is not sufficiently detailed or explained and the claims about results are not adequately quantified or justified. There are several sections (regarding the free vortex simulations and the power and loads of the turbine) of the paper that provide no meaningful input and should be removed entirely.

**Comments:**

Frederik et al. (2020); Munters and Meyers (2018); Frederik et al.(2019))

Please order references by year published (older to newer) and by last name of the first author.

qualitatively validate faster wake breakdown

Why qualitative? The abstract has an adequately defined and quantifiable metric for wake recovery.

at the tip

Figure 5: Installed Gurney flaps (Photographed byTNO)

it is difficult to see the actual Gurney flaps in this picture due to low contrast with the rest of the trailing edge. It may be more useful to show an airfoil cross-section with the geometry of the flaps.

Thus, because of this high uncertainty, the power and loads analysis of field tests is not presentedhere (interested readers may refer Dangi (2023)). estimate the power and loads*

This seems like an important aspect of the study.

Blue

black?

The brown lines in Figure 7 indicate two wind direction limits which capture the prevailing wind sector at the test site. TheScanning LiDAR was placed ≈912m upstream of the wind turbine, and the scan settings utilised are shown in Table 1. Theazimuthal (ψ) ranges of these settings are depicted as red lines in Figure 7. The red center line represents the approximate axisof the wind turbine (for 230◦ wind direction with zero yaw misalignment), and the scan pattern was made symmetric aboutit. The scanning LiDAR azimuth (ψ) and elevation (θ) convention is shown in Figure 8. LOS implies the line of sight of the140LiDAR. The scan time was ≈2.8 minutes, this implies that in a ten minute interval, roughly three samples were available atevery point of the scan.

This section is difficult to follow. Along with the information provided in Table 1, I take it that the scanning lidar was collecting line of site wind speed in a large volume that included the turbine and the wake. The scan is relatively highly resolved in space, but the revisit time is 2.8 minutes, which much contribute to the uncertainty in the description of wake turbulence as information is smeared in time. Is this correct? It would be extremely helpful to have a plan view of the experiment and a perspective drawing of the lidar, scan geometry, and turbine.

he steps takenfor the data processing of the scanning LiDAR data are shown in Figure 9. Firstly, a carrier to noise ratio filter was used suchthat data within the range of −23dB and −3dB was preserved (Cassamo et al. (2021))

This sort of hard filtering based on CNR may be overly conservative and reject viable data or overly generous and keep outliers or spurious data. The dynamic procedure proposed by Beck and Kuhn (2017) offers a more localized approach to filtering and is more likely to provide better flow estimates.

Beck, H., & Kühn, M. (2017). Dynamic data filtering of long-range Doppler LiDAR wind speed measurements. Remote sensing, 9(6), 561.

Table 1: Scanning LiDAR settings (Wind turbine is located at ≈41.5◦ azimuth)

Very little information is provided as to how this scan pattern was designed or what it was designed to prioritize. It is especially important with scanning lidars to carefully assess the goals of the observations against the limitations of the lidars themselves (i.e., what makes these the ideal scans for this study?). Please see more in Letizia, Zhan, and Iungo (2021) and related works.

Letizia, S., Zhan, L., & Iungo, G. V. (2021). LiSBOA (LiDAR Statistical Barnes Objective Analysis) for optimal design of lidar scans and retrieval of wind statistics–Part 1: Theoretical framework. Atmospheric Measurement Techniques, 14(3), 2065-2093.

With the filtered data se

This reads like a thesis, rather than the methods section of a peer-reviewed journal article. I recommend reducing the parts of the methods that are considered common practice and rely on citations instead.

The standard error was defined as

The subscript 'samples' can be removed for the sake of clarity. It is defined in the text in the same sentence.

Figure 10: I

Please increase the font size for figures so that they are easily readable. It is difficult to interpret these graphics due to their size.

How many observations go into each of the curves shown in the figure? Are the shaded regions error or measurement uncertainty? Are the dashed lines the hub height of the turbine?

What should we as readers of your work take from these figures? Perhaps it would be helpful to pull out distributions of the hub-height wind speed and turbulence intensity from the measurements to help readers understand the variability in the sample.

Betz limit of roughly 67%.

The Betz limit is approximately 59.3%. See https://en.wikipedia.org/wiki/Betz%27s_law. Does the figure quoted in the text refer to the limit on velocity deficit? Please review and update.

Figure 12:

This figure appears to show velocity deficit. Please update the caption and use throughout the text for consistency.

assess the impact on the retrofitted wind turbine performance

This is not undertaken in the current study.

4.2 Power and Loads Analysis in Various Inflow Conditions

This manuscript effectively omits any information about power and loads. There are new substantial correlations made between the alleged changes in wind, turbine week, aerodynamics and variations in power and loads. Without the state, it is impossible to say, whether the addition of Gurney flaps has any real purpose or meaningful affect on a winter vine. It is not sufficient to say that a change in momentum deficit is enough to justify their presence.

The increased wake recovery is observed in proximity to the rotor in this wind speed bin as well. It should be noted that the285number of scans for the retrofitted configuration are only 6, which means wake measurements during two 10 minute intervalswere used

Example of six observations is sufficient to quantify average statistics, or measuring uncertainty. Without the presence of some indication of variability. Due to the nature of the skin design, the standard error is not an appropriate metric for uncertainty as it does not factor in the temporal and spatial averaging included in the lidar returns.

The sharp profile of the segmented Gurney flaps designed as shown in Figure 4 was expected to cause an increase innoise level

Does this refer to aeroacoustic noise? If not quantitatively assessed in this work, I recommend moving this to a discussion section about other possible impacts that arise from the Gurney flaps.

Overall, great insights into the wind turbine wake were made possiblewith the field tests conducted in this study. A reduction of the velocity deficit of 10% at 5D downstream distance translatesinto a 30% reduced wake loss

This section appears to be purely conjecture, based on theoretical relationships between wind speed and power production. Without the measurements to support these claims, including wake loss mitigation on a downstream turbine, this sort of discussion should be omitted.

The free vortex wake simulations

This is a purely qualitative description of the model outputs and are difficult to reconcile with the limited measurements from the lidar provided above. No effort is made to describe the details of the simulation setup, the inflow conditions, or the state of the turbine, so it is impossible to infer whether the simulation results pictured in Figure 18 even represent the same case.

---

## Author Comment (AC1)

Dear Anonymous Referee #1,

Thank you for your comments. We provide our rebuttal to the points raised. Below are the author rebuttals in red and anonymous referee #1 comments in blue.

The authors present field data from a wind turbine with and without retrofitted Gurney flaps near the blade tips. The experimental data is velocity data obtained through LIDAR. One section that deals with a power and load analysis based on blade element momentum simulations is added at the end.
We want to kindly mention that the manuscript also contains some more topics which are relevant for a better interpretation of the results and conclusions. The following list of points could be acknowledged for a better summary to the readers of the review:
1. The use of Gaussian process regression in the post processing chain of scanning LiDAR measurements.
2. The analysis of different wind speed retrieval methods.
3. The atmospheric stability implications made.
4. The results of free vortex wake simulations as a validation technique.

I find the paper overall difficult to follow and not well written. The paper is full of grammatical and spelling errors.
We regret that and we have taken the point into account. We will improve the readability in the revised manuscript.

Figures are not cited in order of appearance. Variables in equations are not introduced. References are cited as a list without summarizing their main contributions and relevance to the current manuscript. Figures lack axis tick labels.
Figure citing will be updated in the revised version, as well as introduction of variables used. To the author's interpretation, the relevance of the cited research is clearly defined in the introduction.

Only Figures 1 and 6 miss axis tick labels. This is because of confidentiality reasons as highlighted in the 'Code and Data Availability section'. We want to mention that we have properly labelled all the other figures in the manuscript.

The term 'segmented Gurney flaps' is used before explaining what it means, and then on page 3, the authors suddenly start to talk about miniature trailing edge effectors instead.
We agree that the segmented aspect is not clearly defined early on in the manuscript, and will provide more clarity in the revised manuscript. As the term miniature trailing edge effectors (used in cited literature) is perceived as confusing, we will replace it with segmented Gurney flaps in the revision.

The results themselves are not convincing. The authors themselves seem to be torn. They go from admitting that the impact of Gurney flaps remains within the measurement uncertainty band in the abstract to claiming that great insights into the wind turbine wake were obtained in various conditions.

Unfortunately, this point is a serious misinterpretation of our statements. We talk about the results being within the measurement uncertainty for the 'power and loads' analysis, while the great insights remark is made for the 'wake analysis' conducted in this study. The coupling of these two leads to a major misinterpretation of our results and conclusions.

In my opinion, the data shown does not allow any conclusion to be drawn. Too much information is missing.

Could you let us know what information is missing? Additionally, the 'Code and Data Availability' section states that wake analysis data is available upon request. Also note that to keep the manuscript concise we try to only include the most relevant information for repeatability. The thesis report (cited on line 125) provides more background information.

The authors refer to the standard deviation of the data as the standard error. A distinction between data fluctuations and measurement errors is not made.

Could you further elaborate on the point? We have clearly defined the standard error differently to standard deviation, on line 229.

As evident by various existing literature sources, which are also cited in the paper, standard error is the metric which is typically used for scanning LiDAR measurements; which is implemented in the results of this study as well.

The wake deficit is presented in the form of a spanwise average. I do not understand why the authors do this, as the average value depends strongly on the spatial sampling distribution and the area over which they average. There is too much bias.

Please note that both local (50% span) profiles (vertical) and spanwise average profiles (axial) are provided in the manuscript. The spanwise average metric is also explained in the manuscript (lines 211 to 216), including how the area is chosen.

By introducing the various checks in the data set, relating to inflow conditions and wind turbine performance (see also lines 170 to 173), the comparison is made unbiased to the best extent possible. In addition to that the standard error is clearly indicated in the plots.

The authors hope that these points can be taken into account as well for a second review.

---

## Author Comment (AC2)

Dear Anonymous Referee #2,

Thank you very much for your comments.

We believe that your major comment is about the power and loads analysis of the
retrofitted wind turbine. The current manuscript presents the quantitative power
and loads analysis results through simulations. We will expand this section by
incorporating the power and loads analysis through field tests as well. However, it i's
important to note that this addition does not impact the results of the wake analysis
presented in the manuscript, which is considered as the main result. The post-
processing chain ensured a robust comparison between the wake of the baseline
and retrofitted wind turbines, as detailed in lines 169 through 173.

Next, we believe that the comments about uncertainty quantification and CNR
thresholds are justified with existing literature and are also further explained in this
document below.

We will revise the manuscript to address your above-mentioned comments and the
other minor specific comments. Below are the author responses in red and
anonymous referee #2 comments in blue.

Replies to specific comments:

The manuscript entitled, "Segmented Gurney Flaps for Enhanced Wind Turbine
Wake Recovery" by Nirav Dangi, Koen Boorsma, Edwin Bot, Wim Bierbooms, and
Wei Yu endeavors to describe differences in wind, turbine week aerodynamics
introduced by the presence of a gurney flaps. On the trailing edges of wind turbine
Blade tips. There is a long history of flaps as passive or active control, mechanisms
for wind, turbines, and a great deal of debate in the literature as to their merit. The
authors are strongly encouraged to review this literature and contextualize their
work within the spotty before presenting this for publication again.

We acknowledge that, however, please note that we are not focussing on studies
which use plain flaps for active or passive control. Instead, we  focus on  the use of
Gurney flaps mainly with the purpose to influence wake breakdown, and references
studying this aspect are included in the manuscript.

Overall the manuscript reads like a section of a thesis. By itself, the study is not sufficiently detailed or explained and the claims about results are not adequately quantified or justified. There are several sections (regarding the free vortex simulations and the power and loads of the turbine) of the paper that provide no meaningful input and should be removed entirely.

Perhaps it would help if the statement 'reads like a section of a thesis' is clarified, i.e. what is meant exactly (e.g. wording, structure). We will improve the readability and add the details which we did not provide in the manuscript. The power and loads analysis will be expanded.  The free vortex wake simulations will be omitted , and referral will be made to the corresponding project report, for the interested reader.

Please order references by year published (older to newer) and by last name of the first author.

Noted, we will make the corresponding change.

Why qualitative? The abstract has an adequately defined and quantifiable metric for wake recovery.

Noted, we will reword the statement.

it is difficult to see the actual Gurney flaps in this picture due to low contrast with the rest of the trailing edge. It may be more useful to show an airfoil cross-section with the geometry of the flaps.

Right, we will incorporate that.

This seems like an important aspect of the study.

Noted; for the field tests' power and load analysis we previously referred to the corresponding project report, but now we will include it in the revised manuscript.

black?

Apologies for the confusion, we will reword it as 'black arc' within the brown lines.

This section is difficult to follow. Along with the information provided in Table 1, I take it that the scanning lidar was collecting line of site wind speed in a large volume that included the turbine and the wake. The scan is relatively highly resolved in space, but the revisit time is 2.8 minutes, which much contribute to the uncertainty in the description of wake turbulence as information is smeared in time. Is this correct? It would be extremely helpful to have a plan view of the experiment and a perspective drawing of the lidar, scan geometry, and turbine.

Noted, we will reword it. That is indeed correct.

Please also note that the Figures in Appendix A provide a detailed view of the scan
pattern in cartesian co-ordinates and also a side view which presents the pattern in
relation to the wind turbine.

This sort of hard filtering based on CNR may be overly conservative and reject viable
data or overly generous and keep outliers or spurious data. The dynamic procedure
proposed by Beck and Kuhn (2017) offers a more localized approach  to filtering and
is more likely to provide better flow estimates.

Thank you for the suggestion- this is out of scope of the study at this stage, and will
be added as a recommendation for future work.  Previous studies which incorporate
the hard CNR filter were included in the manuscript. Below is another reference for
the same:

1.  Bodini, N., Zardi, D., & Lundquist, J. K. (2017). Three-dimensional structure of
wind turbine wakes as measured by scanning lidar. Atmospheric
Measurement Techniques, 10(8), 2881-2896. -Section 3.1

Nevertheless, we believe in the robustness of the results with the post processing
chain used. The standard error is indicated below for the hub height for one of the
results (Figure 11a). Overall the standard error is quite low and as expected, typically
around the wake edges it is higher because of probe volume averaging in the region
of wake interaction with the freestream.

[Figure]

Very little information is provided as to how this scan pattern was designed or what
it was designed to prioritize. It is especially important with scanning lidars to carefully
assess the goals of the observations against the limitations of the lidars themselves
(i.e., what makes these the ideal scans for this study?). Please see more in Letizia,
Zhan, and Iungo (2021) and related works.

Thank you for the reference. Please note the Figures that were provided in Appendix
A. We will  now also expand the section to indicate the priority of the scan pattern.

This reads like a thesis, rather than the methods section of a peer-reviewed journal
article. I recommend reducing the parts of the methods that are considered common
practice and rely on citations instead.

Noted, we will reword it accordingly.

The subscript 'samples' can be removed for the sake of clarity. It is defined in the text
in the same sentence.

Noted, it will be removed.

Please increase the font size for figures so that they are easily readable. It is difficult
to interpret these graphics due to their size.

Apologies for the same, the sizes will be increased.

How many observations go into each of the curves shown in the figure? Are the
shaded regions error or measurement uncertainty? Are the dashed lines the hub
height of the turbine? What should we as readers of your work take from these
figures? Perhaps it would be helpful to pull out distributions of the hub-height wind
speed and turbulence intensity from the measurements to help readers understand
the variability in the sample.

The number of observations in each is linked to the number of scans indicated in the
Figures 11 and 15. We will now provide the values for the inflow profiles as well.

Noted, we will now provide the absolute values of the hub height inflow metrics.

Throughout the manuscript, the shaded regions are represent by standard errors,
specifically, 1 times the standard error on each side. This will now  be made more
clear by giving the figures extended captions.

The Betz limit is approximately 59.3%. Does the figure quoted in the text refer to the
limit on velocity deficit? Please review and update.

Yes, we refer to the limit on the velocity deficit. We will indicate the same to avoid
confusion.

This figure appears to show velocity deficit. Please update the caption and use
throughout the text for consistency.

Noted, relevant changes will  be made.

This is not undertaken in the current study. This manuscript effectively omits any
information about power and loads.

As mentioned above, the field tests' power and loads analysis of the retrofitted
turbine will be included in the revised manuscript. The simulation results for power and loads analysis of the retrofitted which were provided in the manuscript, will be
expanded as well.

There are new substantial correlations made between the alleged changes in wind,
turbine week, aerodynamics and variations in power and loads. Without the state, it
is impossible to say, whether the addition of Gurney flaps has any real purpose or
meaningful affect on a winter vine. It is not sufficient to say that a change in
momentum deficit is enough to justify their presence.

We are not entirely clear about this comment. So, we would like to clarify that the
main focus of the study was to prove the use of segmented Gurney flaps for faster
wake recovery, thus, have a meaningful impact on the downstream wind turbine,
which is highlighted by the results. In regard to the upstream turbine, we assess the
power and loads, the results of which, as indicated in the manuscript, highlight that
the upstream turbine is not affected considerably.

Example of six observations is sufficient to quantify average statistics, or measuring
uncertainty. Without the presence of some indication of variability. Due to the nature
of the skin design, the standard error is not an appropriate metric for uncertainty as
it does not factor in the temporal and spatial averaging included in the lidar returns.

Noted, we will not mention it as a limitation now. As mentioned to referee #1 as well,
we have seen that the standard deviation or error is used frequently to present the
scanning LiDAR results. See below list of references:

1. Aitken, M. L., R. M. Banta, Y. L. Pichugina, and J. K. Lundquist, 2014:
   Quantifying Wind Turbine Wake Characteristics from Scanning Remote
   Sensor Data. J. Atmos.   Oceanic Technol., 31, 765–787,
   https://doi.org/10.1175/JTECH-D-13-00104.1 - Figure 18, for example
2. Krishnamurthy, R., Reuder, J., Svardal, B., Fernando, H. J. S., & Jakobsen, J. B. (2017).
   Offshore wind turbine wake characteristics using scanning Doppler lidar. Energy
   Procedia, 137, 428-442. – Figure 9, 12, for example
3. Baker, R. W., & Walker, S. N. (1984). Wake measurements behind a large horizontal
   axis wind turbine generator. Solar Energy, 33(1), 5-12. Figure 9, 10, 11, for example

Nevertheless, we will add a recommendation which states about further investigation
into sources of uncertainty.

Does this refer to aeroacoustic noise? If not quantitatively assessed in this work, I
recommend moving this to a discussion section about other possible impacts that
arise from the Gurney flaps.

Yes, it does. Noted, we will move it.

This section appears to be purely conjecture, based on theoretical relationships
between wind speed and power production. Without the measurements to support these claims, including wake loss mitigation on a downstream turbine, this sort of
discussion should be omitted.

Noted, we will reword it: "The results of this study indicate a reduction of the spanwise
averaged velocity deficit by 10%, at 5D downstream. This enhanced wake recovery
was seen when the upstream wind turbine, in free stream conditions, was retrofitted
with segmented Gurney flaps. Such application of segmented Gurney flaps on wind
turbines in the outer rows of a wind farm could potentially enable closer wind turbine
spacing, yielding a higher energy density. Dedicated farm simulations are
recommended to investigate this and further confirm the promising potential of
segmented Gurney flaps".

This is a purely qualitative description of the model outputs and are difficult to
reconcile with the limited measurements from the lidar provided above. No effort is
made to describe the details of the simulation setup, the inflow conditions, or the
state of the turbine, so it is impossible to infer whether the simulation results pictured
in Figure 18 even represent the same case.

Indeed, it was the same set up as the 3.8MW research wind turbine tested on field.
Apologies for not providing adequate information about the free vortex wake
simulations. We have now removed this section as per your suggestion in the
beginning, and only referred to the detailed setup in the corresponding project
report, for the interested reader.